# Improving object detection quality with structural constraints

**Zihao Rong**, **Shaofan Wang** \*, **Dehui Kong**, **Baocai Yin**

Beijing Key Laboratory of Multimedia and Intelligent Software Technology, Beijing Institute of Artificial Intelligence, Faculty of Information Technology, Beijing University of Technology, Beijing, China

\* wangshaofan@bjut.edu.cn

## Abstract

Recent researches revealed object detection networks using the simple "classification loss + localization loss" training objective are not effectively optimized in many cases, while providing additional constraints on network features could effectively improve object detection quality. Specifically, some works used constraints on training sample relations to successfully learn discriminative network features. Based on these observations, we propose Structural Constraint for improving object detection quality. Structural constraint supervises feature learning in classification and localization network branches with Fisher Loss and Equi-proportion Loss respectively, by requiring feature similarities of training sample pairs to be consistent with corresponding ground truth label similarities. Structural constraint could be applied to all object detection network architectures with the assist of our Proxy Feature design. Our experiment results showed that structural constraint mechanism is able to optimize object class instances' distribution in network feature space, and consequently detection results. Evaluations on `MSCOCO2017` and `KITTI` datasets showed that our structural constraint mechanism is able to assist baseline networks to outperform modern counterpart detectors in terms of object detection quality.

**Data Availability Statement:** The codes could be downloaded from https://osf.io/suzhd/ MSCOCO2017 dataset could be downloaded from https://cocodataset.org/#download We used a KITTI 2D object dataset converted into PASCAL VOC format, and this converted version is held at

## 1 Introduction

Object detection is a fundamental computer vision technology with a broad range of application scenarios, such as autonomous driving. It's a compound task of object classification and localization. Modern object detectors are trained by matching their detection results with ground truth labels, and then minimizing the loss measuring the differences of these label-prediction matches. Each match's loss is constituted with two terms, measuring classification error and localization error respectively. The complete loss is the sum of the two terms of all matches. In such a loss, each detection result is evaluated independently and only required to fit to the matched ground truth label. Though this loss form is simple, recent researches revealed that object detection networks could not be effectively trained by directly minimizing such a loss in many cases [1], while some researches showed that object detection quality could be effectively improved with additional constraints on intermediate network features [2]. Specifically, recent researches on network-based clustering [3] showed that feature learning could

https://www.kaggle.com/zihaorong/kitti-in-voc-format.

**Funding:** This study was funded by the National Natural Science Foundation of China (https://www.nsfc.gov.cn) in the form of a grant [62172022] and by the Beijing Natural Science Foundation in the form of funds to DK. This study was also funded by the National Natural Science Foundation of China in the form of grants to BY [U1811463, U19B2039]. The funders had no role in study design, data collection and analysis, decision to publish, or preparation of the manuscript.

**Competing interests:** The authors have declared that no competing interests exist.

be effectively guided for the benefit of the main task, under constraints on mutual relations between training samples. This indicates it's possible to optimize object class distributions in network feature space for the benefit of object recognition. Thus, it's reasonable to expect object detection quality improvement by complementing the basic loss form of modern detectors with additional constraints on training sample relations in intermediate feature space.

This work presents a training-sample-relation-based constraint on object detection network training for improving detection quality. We name these *Structural Constraints*, because these constraints exert influence on the structure of training sample distribution in object detection network feature space (as is shown later in Fig 3). Structural constraints append two terms to the basic loss, *Fisher Loss* and *Equi-proportion Loss*, for constraining the relations of training samples in classification branch space and localization branch space respectively. For an arbitrary pair of training samples, Fisher loss measures the difference between pairwise sample feature similarity and pairwise classification target similarity, while equi-proportion loss measures the difference between pairwise sample feature similarity and pairwise localization target similarity. Thus, under the constraint of these two terms, training sample feature distributions in classification branch and localization branch more resemble ground truth label distributions. As a result, features of these network branches could be more easily transformed to accurate detection results. Structural constraints could be applied to object detection networks of various architectures, like single-stage, two-stage and multi-stage networks, without changing the original network structures or influencing detection rates. In our experiments, we evaluated structural constraints' effect on representative object detection networks of various architectures on different image datasets. These experiment results demonstrated that structural constraints could improve object detection quality noticeably on a broad range of detectors.

To summarize, novel contributions of this work are:

- proposing Fisher loss function as part of structural constraints to constrain training sample feature relations for improving classification performance of object detection networks;

- proposing equi-proportion loss function as part of structural constraints to constrain training sample feature relations for improving localization performance;

- a mechanism for applying structural constraints to various object detection network architectures.

The rest of this paper is organized as follows: Section 2 reviews related works, Section 3 describes in detail structural constraint and the mechanism of applying it to networks, Section 4 presents our experiment results and analysis, and finally Section 5 concludes this work.

## 2 Related works

In this section, we review some previous works closely related to structural constraints proposed in this work, and we confine the scope to works based on neural networks. At first, we review some deep learning models for image recognition with feature learning constraints; then, we review representative object detection networks of various architectures.

### 2.1 Feature learning constraints

Feature learning constraints are widely adopted in deep-learning-based image recognition domain. Some works on object detection use feature learning constraints to improve object detection quality. RIFD-CNN [2] used two types of constraints on its network's intermediate layer features, one for rotation invariance and one for Fisher discrimination. Its rotation

invariance constraint requires the intermediate feature representation of each training sample image to be similar to the average intermediate feature representation of the rotated versions of the image, so the subsequent classification based on this type of features will be robust against influence of object rotation. Its Fisher discrimination constraint requires each class's training sample intermediate features to lie close to the mean of the class, and each class's mean feature to lie distant from the global mean of all classes, so the subsequent classification layer could easily and accurately separate the classes from each other. Using these two constraints, RIFD-CNN achieved significant object detection accuracy improvement.

DETR [4] is another object detection network using feature learning constraints. DETR uses transformer to process feature maps from its backbone into detection results. Its transformer's encoder consists of multiple layers of attention mechanism, and the detection results are produced by the last attention mechanism layer. However, other attention mechanism layers' intermediate features are also required to be transformed into accurate detection results through the same detection head shared with the last layer. This deep supervision is in essence a type of feature learning constraint: the supervision on the intermediate attention mechanism layers constrains their output features to facilitate the subsequent inference for better detection accuracy.

Feature learning constraints have also been used to solve image clustering problems. Deep Self-evolution Clustering (DSEC) [3] network and Deep Adaptive Clustering (DAC) [5] network constrain their output features' pairwise relationships to make these features directly express cluster identities. These clustering networks' constraints require the dot products of aribitrary pairs of output features to be close to corresponding pseudo labels. These pseudo labels reflect cosine similarities of the feature pairs. As a result, the training of these networks under this type of constraints gradually makes the output features to be one-hot vectors which express cluster identities directly. Factually, this type of constraints on pairwise feature relationships are the only content in these two clustering networks' training objective functions.

Compared with the feature learning constraints in the works described above, structural constraints in this work exhibit both similarity and difference. Like RIFD-CNN, structural constraints are applied over intermediate layer features of object detection networks; like DSEC and DAC, structural constraints are based on pairwise training sample feature relations. However, the combination of these two characteristics is absent in all these works. Besides, as constraints for object detection networks, RIFD-CNN's constraints are applied for classification merely, while structural constraints are applied for both classification and localization. Furthermore, RIFD-CNN did not constrain inter-class training sample relations, while our structural constraints' Fisher loss constrains both inter- and intra-class relations over all training sample pairs.

## 2.2 Object detection network architectures

Until now, object detection networks exhibited two types of architectures: networks generating detections in a single stage, and networks generating detections through several stages of refinements. We review these architectures below.

**2.2.1 Single-stage object detection networks.** Single-stage object detection networks transform input images' backbone feature maps into detection results directly, through a single detection head. SSD [6] is the forerunner of this architecture. It scatters boxes of various sizes and aspect ratios over input images' feature maps and infers classes and adjustments for these boxes to form detection results. Detection results across feature map pyramid levels are synthesized to infer final detection results. The boxes initially scattered are then known as *anchors*.

YOLO [7] is another single-stage detection network that is fast at inference. It additionally infers a confidence value for each bounding box, which represents probability of existence of

objects within the bounding box, and these confidence values participate in the decision of final detection results. However, YOLO's detection quality is not satisfying.

RetinaNet [1] is a high-detection-quality single-stage object detection network. It focuses on dealing with imbalance between foreground and background training samples, which is a crucial cause of poor detection quality of many other single-stage networks. It proposes Focal Loss to replace the widely adopted cross entropy loss for classification task. By using focal loss, RetinaNet is able to allocate more weights on poorly classified hard samples during training, and make the trained network better generalize to test data.

**2.2.2 Object detection networks with several stages.** Another kind of object detection networks are constituted with more than one stage. These networks could be further divided into two groups according to number of network stages: two-stage networks and multi-stage (more than two) networks. The first network stage of all these object detection networks are responsible for generating region proposals, also known as RoIs (regions of interest). Two-stage networks then refine the region proposals with a detection head to produce final detection results, while multi-stage networks refine the region proposals with several detection heads in sequence. We review representatives of these architectures below.

*Two-stage object detection networks*. Two-stage object detection networks appeared early among all architectures, and usually produce better detection quality than single-stage networks. Faster RCNN [8] is the forerunner of this architecture. Faster RCNN introduced RPN (Region Proposal Network) upon the basis of Fast RCNN [9]. RPN takes backbone feature maps as input and inferences RoIs and corresponding confidence values. These RoIs are then used to extract features from backbone feature maps through RoI pooling operation, and these features are passed into a fully connected detection head to inference detection results.

R-FCN [10] focuses on accelerating inference rate of Faster RCNN by reducing redundant computation of detection head. R-FCN's detection head is constituted with convolutional layers, and is able to generate a special feature map of which different channels are sensitive to different parts of target objects. Then, RoI pooling over this feature map could easily decide whether an RoI accurately localizes an object and corresponding class, by filling RoI parts with features from corresponding channels. Since most necessary computation is done by the convolutional detection head and the remaining RoI pooling operations cost only subtle computation, R-FCN's inference is time-efficient.

Double-head RCNN [11] is another two-stage network whose second stage is composed of two detection heads in parallel, one fully connected head and one convolutional head. This design is based on the observation that fully connected layers are sensitive to spatial completeness of objects, while convolutional layers are robust against occlusion and deformation. Thus Double-head RCNN uses its fully connected head to infer classification scores which should reflect localization quality, and uses its convolutional head to infer bounding boxes to better generalize to various object appearances and influencing contents.

*Multi-stage object detection networks*. Multi-stage object detection networks extend two-stage architecture by appending additional detection heads, refining RoIs with more stages of inferences. Cascade RCNN [12] is a typical multi-stage object detection network. During Cascade RCNN training, each stage's detection head is trained from detection results of its previous stage. At inference, each stage's detection head takes features from RoI pooling based on its previous stage's detection boxes, and generates new detection results. The final detection results take the last stage's detection head's output boxes as localization results, and take the averages of all detection heads' class scores as classification results. The increased network stages improved detection quality noticeably, making Cascade RCNN one of the most accurate object detectors by then.

Hybrid Task Cascade [13] is a multi-stage network capable of both object detection and instance segmentation. Hybrid Task Cascade inherited the network structure of Cascade RCNN, and introduced additional components and links. It introduced a semantic segmentation convolutional branch to provide helpful inputs to its detection heads and mask heads. The detection quality of Hybrid Task Cascade is outstanding in multi-stage group, but the whole of its network is cumbersome.

All representative object detection networks mentioned above and many others lack constraints on relationships between training samples in feature spaces, so structural constraints proposed in this work are able to complement them in this respect. We will show that structural constraints are applicable to all these architectures through a unified mechanism in next section.

## 3 Structural constraint mechanism

In this section, we describe sturctural constraint mechanism for object detection in detail. Firstly, we explain the motivation of structural constraints. Then, we present the definition of structural constraints. After these, we describe the mechanism of combining structural constraints with object detection networks.

### 3.1 Motivation

The reason of we proposing structural constraints is based on two observations: first, the lack of constraints on training sample relationships in modern object detection networks; second, the importance of feature learning exhibited in many other image recognition tasks. As described in Section 1, it could be observed that most modern object detection networks' loss functions usually have a form like this:

$$\sum_i L_{cls}(p_i, p_i^{gt}) + L_{loc}(b_i, b_i^{gt}) \tag{1}$$

where $L_{cls}$ and $L_{loc}$ are two loss terms for measuring classification error and localization error respectively. For each match, the difference between the estimated class probability vector $p_i$ and the corresponding ground truth vector $p_i^{gt}$ is calculated by $L_{cls}$, and the difference between the estimated bounding box $b_i$ and the corresponding ground truth box $b_i^{gt}$ is measured by $L_{loc}$. Loss functions like this only force each detection result to fit to its matched ground truth. They are simple in form, but could not be effectively minimized in many cases, since the supervision on object classification could be severely influenced by large amount of background training samples [1]. We observed that additional supervision on one training sample could come from the other training samples, since one training sample could be represented by its relative differences from the others. This could be understood by looking at some works on image clustering, such as DSEC [3], where the clustering network was effectively trained under the supervision on similarity of each pair of training samples. Thus, structural constraints are designed to supervise the differences between each pair of sample detections. Because of that object detection consists of classification and localization, structural constraints use two types of loss functions to measure sample pairs' classification differences and localization differences, namely Fisher loss and equi-proportion loss.

We also observed that proper supervision on object detection networks' intermediate features could effectively improve detection quality. Examples are RIFD-CNN [2]'s rotation invariance constraint and Fisher discrimination constraint on its backbone's intermediate layers, and DETR [4]'s auxiliary supervisions on multiple levels of transformer decoders. Apart from this, we try to avoid disrupting optimization of the main objective in Eq (1). Thus,

instead of being applied over object detection networks' final outputs, structural constraints are applied over the networks' intermediate features to guide the feature learning.

### 3.2 Definition

Structural constraints take training samples' intermediate features as input. To supervise training samples' relations during classification and localization, structural constraints use Fisher loss and equi-proportion loss to constrain pairwise feature differences respectively. These losses in structural constraints and the basic object detection objective in Eq (1) altogether form the new training objective.

Fisher loss in structural constraints calculates the similarity between an arbitrary pair of intermediate features of training samples, and supervises this with the corresponding pair of class labels' similarity. It's expressed as:

$$L_{\text{Fisher}}(f_i, f_j, p_i^{\text{gt}}, p_j^{\text{gt}}) = [\sigma(f_i) \cdot \sigma(f_j) - p_i^{\text{gt}} \cdot p_j^{\text{gt}}]^2 \tag{2}$$

where $\sigma(\cdot)$ is sigmoid function, $f_i$ is a transformed intermediate feature vector of training sample $i$, and $p_i^{\text{gt}} \in [0, 1]^C$ is the corresponding one-hot class label, with $C$ being the number of object classes. Fisher loss $L_{\text{Fisher}}$ calculates the similarity between $f_i$ and $f_j$, and the similarity between $p_i^{\text{gt}}$ and $p_j^{\text{gt}}$, both in terms of dot production. The squared difference between these two similarities is used as the loss value. To make the comparison between these similarities fair, $f_i$ is obtained by linearly transforming the intermediate feature into the same dimensionality as $p_i^{\text{gt}}$. Since $f_i$ acts as a proxy of the intermediate feature, we name it *Proxy Feature*. Before calculating the similarity, the proxy feature vectors' elements are transformed by $\sigma(\cdot)$ into the same range $[0, 1]$ as $p_i^{\text{gt}}$. By supervising the similarity between proxy feature vectors, Fisher loss drives the similarity between the underlying intermediate features to be consistent with the similarity of the corresponding class labels. As a result, Fisher loss produces the effect of reducing intra-class variance and increasing inter-class separation of training sample distribution, which benefits object classification.

Equi-proportion loss is another loss term in structural constraints. It also measures the similarity between an arbitrary pair of intermediate features, but supervises this with the corresponding pair of localization labels. It's expressed as:

$$L_{\text{equip}}(f_i', f_j', b_i^{\text{gt}}, b_j^{\text{gt}}) = \|\sigma(\frac{\sigma(f_i')}{\sigma(f_j')}) - \sigma(\frac{\sigma(b_i^{\text{gt}})}{\sigma(b_j^{\text{gt}})})\|^2 \tag{3}$$

where $f_i'$ is proxy feature of training sample $i$, and $b_i^{\text{gt}} \in \mathbb{R}^4$ is the corresponding localization label. $f_i'$ is linearly transformed from the intermediate feature into same dimensionality as $b_i^{\text{gt}}$, to facilitate the comparison between training sample difference and localization label difference. Since $b_i^{\text{gt}}, b_j^{\text{gt}}$ are not bounded, we measure their relative difference in terms of element-wise ratios, and so is the difference between $f_i'$ and $f_j'$ measured. The squared magnitude of the difference between these two sets of ratios is used as the value of $L_{\text{equip}}$. Under the guidance of equi-proportion loss, the intermediate features of training samples tend to be sensitive enough to reflect the differences between their localization labels, and benefit bounding box regression.

After applying structural constraint constituted with Fisher loss and equi-proportion loss, the object detection network training objective is rewritten as:

$$\sum_{i}[\mathrm{L}_{\mathrm{cls}}(p_i, p_i^{\mathrm{gt}}) + \mathrm{L}_{\mathrm{loc}}(b_i, b_i^{\mathrm{gt}})]$$

$$+ \sum_{i,j}[\mathrm{L}_{\mathrm{Fisher}}(f_i, f_j, p_i^{\mathrm{gt}}, p_j^{\mathrm{gt}}) + \mathrm{L}_{\mathrm{equip}}(f_i', f_j', b_i^{\mathrm{gt}}, b_j^{\mathrm{gt}})]$$

(4)

where Fisher loss and equi-proportion loss are evaluated for all pairs of training samples. This sum of original loss and structural constraint terms is used to calculate back-propagations during end-to-end object detection network training processes. Thus, training with this new objective not only optimizes the main objective of object detection, but also optimizes the structure of training sample distribution in intermediate feature space which benefits the main objective in return.

## 3.3 Combination with various object detection architectures

Structural constraints supervise intermediate features of object detection networks, that is, applied over intermediate network layers, so how they are combined with networks depends on the forms of these layers, which differ among object detection architectures. We describe how structural constraints are combined with single-stage, two-stage and multi-stage object detection networks respectively below.

**Single-stage case.** Single-stage object detection networks' detection heads transform backbone feature maps with two-dimensional convolution (Conv2D) to generate classification outputs and localization outputs. Because that the dimensionality of proxy features used in Fisher loss and equi-proportion loss calculation must be unified with the dimensionality of classification outputs and localization outputs respectively, structural constraints in single-stage networks use Conv2D layers to transform intermediate features of training samples into the needed proxy features. This could be expressed as:

$$
\begin{aligned}
\{f_i\}_i &= \mathrm{Conv2D}_{\mathrm{Fisher}}(F) \\
\{f_i'\}_i &= \mathrm{Conv2D}_{\mathrm{equip}}(F)
\end{aligned}
$$

(5)

where $\mathrm{Conv2D}_{\mathrm{Fisher}}$ and $\mathrm{Conv2D}_{\mathrm{equip}}$ are convolution layers generating proxy features for Fisher loss and equi-proportion loss respectively, and $F$ is intermediate feature collection. Conv2D$_{\mathrm{Fisher}}$ and Conv2D$_{\mathrm{equip}}$ take $F$ as input and generate proxy feature collections $\{f_i\}_i$, $\{f_i'\}_i$. It should be noticed that $F$, $\{f_i\}_i$ and $\{f_i'\}_i$ take the form of feature tensors in this case. With proxy features obtained, the rest of structural constraint evaluation is exactly same as the description in Section 3.2. The complete mechanism in single-stage case is illustrated in Fig 1a.

**Two-stage case.** Two-stage object detection networks firstly generate RoIs with their RPNs, and then their detection heads infer detection results from these RoIs. Their detection heads usually consist of fully-connected (FC) layers. Thus, for the same reason as in single-stage case, we set up special FC layers for transforming intermediate features into proxy features whose dimensionality is unified with detection head outputs. This could be expressed as:

$$
\begin{aligned}
\{f_i\}_i &= \mathrm{FC}_{\mathrm{Fisher}}(F) \\
\{f_i'\}_i &= \mathrm{FC}_{\mathrm{equip}}(F)
\end{aligned}
$$

(6)

where $\mathrm{FC}_{\mathrm{Fisher}}$ and $\mathrm{FC}_{\mathrm{equip}}$ are the FC layers that generate proxy features for Fisher loss and equi-proportion loss respectively. In this case, the intermediate feature collection $F$ comes

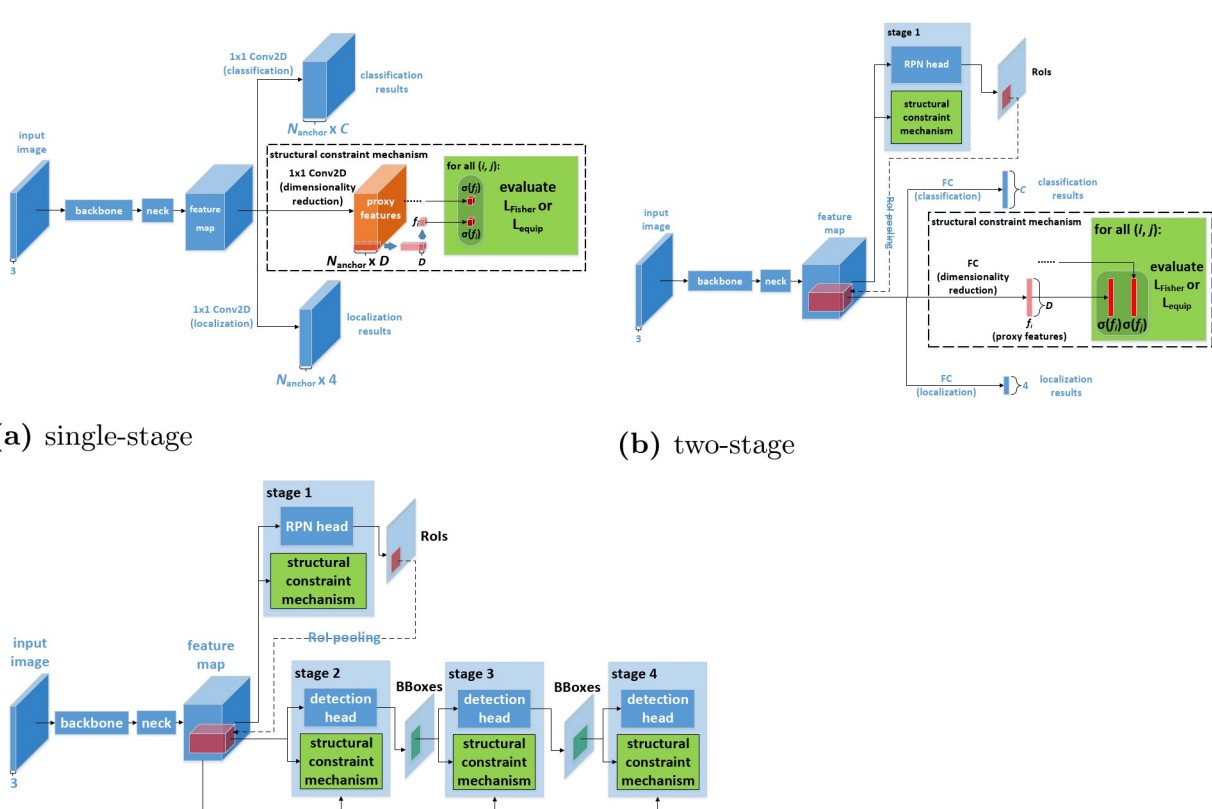

**(a)** single-stage

**(b)** two-stage

**(c)** multi-stage

**Fig 1. Illustration of structural constraint mechanisms in object detection networks of various architectures.** (a) single-stage, (b) two-stage, and (c) multi-stage.

from RoI pooling. The rest of structural constraint evaluation is still same as the description in Section 3.2. Apart from the detection heads, structural constraints could also be applied to RPNs of two-stage networks, because these RPNs are identical to single-stage networks' detection heads. This means the aforementioned mechanism for single-stage case could be directly applied to these RPNs. The complete structural constraint mechanism for two-stage case is illustrated in Fig 1b.

**Multi-stage case.** Multi-stage object detection networks extend two-stage architecture by using multiple detection heads to refine detection results sequentially. Thus, compared with two-stage networks, the constituting modules of multi-stage networks remain unchanged. This means how structural constraints are applied to detection heads and RPNs in multi-stage networks is exactly same as the two-stage case. For structural constraints on detection heads, the proxy features are generated in the same manner as Eq (6); on RPNs, they are generated in the same manner as Eq (5). All the rest of structural constraint evaluation still obey Section 3.2. The complete mechanism for multi-stage case is illustrated in Fig 1c.

In all cases above, structural constraint mechanisms exist during the training period of these object detection networks, and guide the intermediate feature learning by handling proxy features. At inference time, all calculations related to structural constraints are absent, as well as all exclusive network layers (Conv2D$_{Fisher/equip}$, FC$_{Fisher/equip}$), so detection rates and deployment sizes of these networks are not influenced.

## 4 Experiments

To verify the effectiveness of structural constraints, we experimented with multiple object detection networks over several image datasets, and examined the training processes and network behaviors. In this section, we present these experiment results.

### 4.1 Experiment settings

We describe settings of the experiments firstly. These include settings of networks, training and testing. All hyper-parameters listed below are set to default values of MMDetection [14] configuration files.

**Networks.** The default settings of object detection networks used in the experiments are: ResNet-101 [15] as backbone, FPN [16] as neck, and Greedy NMS [17] for post-processing. All multi-stage networks use 3 stages of detection heads. All object detection networks are implemented with MMDetection toolbox [14] and PyTorch deep learning library [18].

**Training and testing.** All networks' optimizers are SGD (Stochastic Gradient Descent). The default length of training is 12 epochs. For single-stage networks, their detection head training samples' positive and negative IoU thresholds are 0.5 and 0.4 respectively. For two-stage networks, their detection head training samples' positive and negative IoU thresholds are both 0.5, and positive training samples cover 25%. For multi-stage networks, 3 stages of detection heads' positive and negative IoU thresholds are 0.5, 0.6 and 0.7 respectively. Besides, for two- and multi-stage networks, their RPN training samples' positive and negative IoU thresholds are 0.7 and 0.3 respectively, and positive samples cover 50%. All training samples are randomly obtained. At test time, the default NMS IoU threshold is 0.5 for detection heads, and 0.7 for RPNs. All networks are trained and tested on GPU servers.

### 4.2 Experiment results

We present experiment results on structural constraint mechanism in this subsection. Firstly, we present ablation evaluation results to show influences of different factors in the mechanism. Then, we compare object detection quality of our structural-constraint-applied networks with other modern detectors. Finally, we analyze behaviors of structural constraint mechanism through visualization.

**4.2.1 Ablation evaluation.** We performed ablation evaluations on structural constraint mechanism to investigate different factors' influences on object detection quality, including the constituting loss terms $L_{Fisher}$ and $L_{equip}$ as well as different combination manners. We report our evaluation results on two widely used image datasets, MSCOCO2017 [19] and KITTI [20], respectively.

*Evaluations on MSCOCO2017.* For ablation on MSCOCO2017, all object detection networks are trained over the train2017 subset, and tested over the val2017 subset. We choose RetinaNet as the evaluation subject for single-stage architecture, Faster RCNN for two-stage, and Cascade RCNN for multi-stage. The ablation evaluation results are shown in Table 1. The network names containing "+$L_{Fisher/equip}$" indicate that Fisher loss or equi-proportion loss is applied to the detection heads of those networks, and names with "+$L_{Fisher}/equip^2$" indicate Fisher loss or equi-proportion loss is applied to both the detection heads and RPNs of those networks (in two- or multi-stage case). It could be observed that structural constraint mechanism is able to improve object detection qualities of all network subjects on this general object detection task. Specifically, the complete structural constraint mechanism that includes both Fisher loss and equi-proportion loss produced the most obvious improvement in some cases, like Faster RCNN + $L_{Fisher}^2$ + $L_{equip}^2$. We also evaluated the

**Table 1. Ablation evaluations of structural constraint mechanism on `MSCOCO2017`.**

| detector | AP | AP$_{0.5}$ | AP$_{0.75}$ | AP$_{small}$ | AP$_{med}$ | AP$_{large}$ | AR$_{MD=1}$ | AR$_{MD=10}$ | AR | AR$_{small}$ | AR$_{med}$ | AR$_{large}$ |
|---|---|---|---|---|---|---|---|---|---|---|---|---|
| RetinaNet | 0.379 | 0.572 | 0.409 | 0.203 | 0.420 | 0.498 | 0.321 | 0.513 | 0.544 | 0.335 | 0.591 | 0.699 |
| RetinaNet+L$_{Fisher}$ | 0.370 | 0.559 | 0.398 | 0.194 | 0.410 | **0.502** | 0.317 | 0.504 | 0.535 | 0.321 | 0.581 | **0.701** |
| Faster RCNN | 0.376 | 0.589 | 0.412 | 0.213 | 0.419 | 0.481 | 0.315 | 0.502 | 0.526 | 0.337 | 0.569 | 0.669 |
| Faster RCNN+L$_{Fisher}$ | **0.377** | 0.589 | **0.413** | 0.213 | 0.414 | **0.485** | 0.315 | 0.500 | 0.524 | 0.329 | **0.571** | 0.657 |
| Faster RCNN + L$_{Fisher}^2$ | **0.378** | 0.588 | 0.410 | **0.217** | 0.419 | **0.488** | **0.317** | **0.505** | **0.530** | 0.335 | **0.575** | **0.673** |
| Faster RCNN* | 0.383 | 0.606 | 0.414 | 0.223 | 0.427 | 0.496 | 0.314 | 0.496 | 0.521 | 0.332 | 0.564 | 0.662 |
| Faster RCNN+L$_{equip}$ * | **0.384** | **0.607** | **0.417** | **0.224** | 0.427 | **0.501** | **0.317** | **0.501** | **0.524** | **0.336** | **0.568** | **0.664** |
| Faster RCNN + L$_{equip}^2$ * | 0.383 | **0.607** | 0.412 | **0.224** | **0.429** | 0.496 | 0.314 | 0.495 | 0.519 | 0.332 | **0.569** | 0.656 |
| Faster RCNN+L$_{Fisher}$+L$_{equip}$ * | **0.384** | **0.607** | **0.415** | **0.227** | 0.426 | **0.500** | 0.314 | **0.498** | **0.522** | **0.335** | **0.567** | 0.661 |
| Faster RCNN + L$_{Fisher}^2$ + L$_{equip}^2$ * | **0.385** | **0.607** | **0.418** | **0.225** | 0.427 | **0.505** | **0.318** | **0.501** | **0.525** | 0.330 | **0.567** | **0.671** |
| Cascade RCNN | 0.412 | 0.590 | 0.447 | 0.227 | 0.447 | 0.550 | 0.337 | 0.530 | 0.554 | 0.337 | 0.595 | 0.718 |
| Cascade RCNN+L$_{Fisher}$ | 0.412 | **0.592** | **0.451** | **0.231** | **0.448** | 0.552 | 0.337 | 0.529 | 0.554 | **0.346** | **0.596** | **0.719** |
| Cascade RCNN + L$_{Fisher}^2$ | 0.412 | **0.592** | **0.449** | **0.233** | **0.449** | 0.545 | 0.337 | **0.531** | 0.554 | **0.341** | **0.599** | 0.706 |
| Cascade RCNN* | 0.396 | 0.570 | 0.433 | 0.218 | 0.428 | 0.524 | 0.332 | 0.523 | 0.546 | 0.334 | 0.585 | 0.700 |
| Cascade RCNN+L$_{equip}$ * | 0.395 | 0.567 | 0.432 | 0.214 | 0.426 | 0.522 | 0.332 | 0.521 | 0.544 | 0.321 | **0.586** | **0.702** |
| Cascade RCNN + L$_{equip}^2$ * | 0.396 | 0.568 | 0.433 | **0.224** | 0.427 | 0.520 | **0.334** | **0.525** | **0.548** | **0.336** | **0.586** | **0.711** |

Note: the values where improvements happen are in **bold face**.

"*" indicates that the network is trained using smaller batch size.

influence of batch size, and the networks marked with "*" are trained with smaller batch sizes (half). It could be observed that structural constraint mechanism is robust against batch size changes.

*Evaluations on `KITTI`.* We use the 2D object detection subset in `KITTI` to perform ablation evaluations, which contains 7481 labeled driver-view images. For all evaluated network subjects, the first 6000 images are used for training and the rest 1481 images for testing. We adopted Pascal-VOC-styled metrics which evaluate class-wise average precisions and the global mean average precision (MAP). We choose RetinaNet and SSD as evaluation subjects for single-stage architecture, Faster RCNN for two-stage, and Cascade RCNN for multi-stage. The evaluation results are shown in Table 2. It could be observed that structural constraint mechanism is able to produce object detection quality improvement for all these network architectures. It's also observable that the improvement happened on multiple classes simultaneously, such as the case of Faster RCNN + L$_{Fisher}^2$. Besides, structural constraint mechanism still exhibits robustness against batch size settings, which could be observed from the evaluations on Cascade RCNN.

**4.2.2 Comparison with other object detectors.** We present object detection quality comparisons between modern object detectors and our networks with structural constraints in this subsection. These comparisons were carried out over `MSCOCO2017` and `KITTI`. We give descriptions respectively in the following.

*Comparison on `MSCOCO2017`.* The training set and testing set for this comparison are same as the settings in last subsection. The evaluation results are presented in Table 3. *SCM-Two* and *SCM-Multi* are our two-stage and multi-stage object detection networks with structural constraint mechanisms. *SCM-Two* is configured as Faster RCNN + L$_{Fisher}^2$ + L$_{equip}^2$, and *SCM-Multi* as Cascade RCNN + L$_{Fisher}^2$. *SSD300* and *SSD512* are SSD networks with input image sizes as 300 × 300 and 512 × 512 respectively. It could be observed that our *SCM-Two*

**Table 2. Ablation evaluations of structural constraint mechanism on KITTI.**

| detector | car | pedestrian | van | truck | person sitting | cyclist | tram | misc | don't care | MAP |
|---|---|---|---|---|---|---|---|---|---|---|
| RetinaNet | 0.977 | 0.925 | 0.989 | 1.00 | 0.927 | 0.985 | 0.997 | 0.966 | 0.828 | 0.955 |
| RetinaNet+$L_{Fisher}$ | 0.977 | 0.902 | 0.987 | 1.00 | **0.942** | 0.976 | 0.997 | **0.969** | 0.805 | 0.950 |
| SSD | 0.856 | 0.395 | 0.685 | 0.826 | 0.231 | 0.408 | 0.806 | 0.509 | 0.123 | 0.538 |
| SSD+$L_{equip}$ | 0.853 | **0.409** | **0.704** | **0.827** | 0.219 | **0.417** | **0.835** | 0.499 | 0.118 | **0.542** |
| SSD+$L_{Fisher}$+$L_{equip}$ | 0.856 | 0.388 | **0.715** | 0.805 | 0.207 | 0.396 | **0.820** | 0.506 | **0.129** | 0.536 |
| Faster RCNN | 0.978 | 0.932 | 0.994 | 1.00 | 0.816 | 0.979 | 1.00 | 0.990 | 0.845 | 0.948 |
| Faster RCNN+$L_{Fisher}$ | **0.979** | 0.928 | **0.996** | 1.00 | **0.874** | **0.995** | 1.00 | 0.990 | 0.844 | **0.956** |
| Faster RCNN + $L_{Fisher}^2$ | **0.979** | 0.932 | **0.996** | 1.00 | **0.884** | 0.986 | 1.00 | 0.985 | **0.849** | **0.957** |
| Cascade RCNN | 0.976 | 0.928 | 0.993 | 1.00 | 0.853 | 0.983 | 1.00 | 0.990 | 0.871 | 0.955 |
| Cascade RCNN+$L_{Fisher}$ | 0.976 | 0.922 | **0.994** | 1.00 | 0.836 | **0.986** | 1.00 | **0.995** | **0.878** | 0.954 |
| Cascade RCNN + $L_{Fisher}^2$ | 0.976 | 0.917 | **0.994** | 1.00 | **0.873** | 0.982 | 0.991 | **0.995** | **0.882** | **0.957** |
| Cascade RCNN* | 0.943 | 0.792 | 0.943 | 0.971 | 0.599 | 0.904 | 0.977 | 0.920 | 0.354 | 0.822 |
| Cascade RCNN+$L_{Fisher}$+$L_{equip}$ * | 0.939 | 0.789 | 0.936 | **0.979** | **0.646** | 0.898 | 0.946 | **0.925** | 0.343 | 0.822 |
| Cascade RCNN + $L_{Fisher}^2$ + $L_{equip}^2$ * | 0.939 | **0.804** | 0.934 | **0.987** | **0.608** | 0.900 | 0.918 | 0.885 | 0.354 | 0.814 |

Note: the values where improvements happen are in **bold face**;

"*" indicates that the network is trained using smaller batch size.

network produced identical object detection quality with many other detectors, and our *SCM-Multi* network achieved top values under most metrics.

*Comparison on* KITTI. In this comparison, the training setting of our network *SCM-Multi* is same as the last subsection, and it's configured as Cascade RCNN + $L_{Fisher}^2$ + $L_{equip}^2$. Other detectors' evaluation results are obtained from KITTI's official website. The comparison is shown in Table 4. Since KITTI's leaderboard publishes detection precisions on car, pedestrian and cyclist, we compare performances on these three classes and the global mean

**Table 3. Object detection quality comparison between structural-constraint-applied networks and other detectors on MSCOCO2017.**

| detector | AP | $AP_{0.5}$ | $AP_{0.75}$ | $AP_{small}$ | $AP_{med}$ | $AP_{large}$ | $AR_{MD=1}$ | $AR_{MD=10}$ | AR | $AR_{small}$ | $AR_{med}$ | $AR_{large}$ |
|---|---|---|---|---|---|---|---|---|---|---|---|---|
| FCOS [21] | 0.391 | 0.585 | 0.418 | 0.220 | 0.435 | 0.511 | - | - | - | - | - | - |
| Mask Scoring RCNN [22] | 0.400 | **0.614** | 0.437 | 0.232 | 0.442 | 0.523 | - | - | - | - | - | - |
| GA-RetinaNet [23] | 0.389 | 0.591 | 0.418 | 0.220 | 0.426 | 0.519 | - | - | - | - | - | - |
| RetinaNet-GHM [24] | 0.390 | 0.577 | 0.413 | 0.218 | 0.432 | 0.518 | - | - | - | - | - | - |
| Libra Faster RCNN [25] | 0.403 | 0.612 | 0.439 | 0.233 | 0.443 | 0.522 | - | - | - | - | - | - |
| SSD300 [6] | 0.254 | 0.428 | 0.264 | 0.059 | 0.279 | 0.428 | 0.238 | 0.348 | 0.368 | 0.094 | 0.413 | 0.588 |
| SSD512 [6] | 0.292 | 0.481 | 0.307 | 0.105 | 0.347 | 0.456 | 0.262 | 0.392 | 0.415 | 0.138 | 0.492 | 0.614 |
| Mask RCNN [26] | 0.387 | 0.597 | 0.424 | 0.226 | 0.427 | 0.501 | 0.322 | 0.512 | 0.537 | 0.349 | 0.582 | 0.674 |
| Double-head RCNN [11] | 0.386 | 0.583 | 0.420 | 0.225 | 0.422 | 0.496 | 0.326 | 0.522 | 0.549 | **0.350** | 0.590 | 0.700 |
| DETR [4] | 0.401 | 0.606 | 0.420 | 0.183 | 0.433 | **0.595** | - | - | - | - | - | - |
| YOLOX [27] | 0.403 | 0.591 | 0.434 | **0.235** | 0.445 | 0.531 | - | - | - | - | - | - |
| Dynamic R-CNN [28] | 0.389 | 0.576 | 0.427 | 0.221 | 0.419 | 0.517 | - | - | - | - | - | - |
| SCM-Two (ours) | 0.385 | 0.607 | 0.418 | 0.225 | 0.427 | 0.505 | 0.318 | 0.501 | 0.525 | 0.330 | 0.567 | 0.671 |
| SCM-Multi (ours) | **0.412** | 0.592 | **0.449** | 0.233 | **0.449** | 0.545 | **0.337** | **0.531** | **0.554** | 0.341 | **0.599** | **0.706** |

Note: the top value under each metric is in **bold face**.

**Table 4. Object detection quality comparison of our structural-constraint-applied networks and other detectors on `KITTI`.**

| detector | car | pedestrian | cyclist | MAP |
|---|---|---|---|---|
| TuSimple [29] | 0.908 | 0.770 | 0.814 | 0.831 |
| RRC [30] | 0.906 | 0.753 | 0.850 | 0.836 |
| UberATG-MMF [31] | 0.918 | - | - | - |
| PC-CNN-V2 [32] | 0.908 | - | - | - |
| SJTU-HW [33] | 0.908 | 0.742 | - | - |
| SenseKITTI [34] | 0.908 | 0.673 | 0.818 | 0.800 |
| F-PointNet [35] | 0.908 | 0.773 | 0.849 | 0.843 |
| HRI-VoxelFPN [36] | 0.907 | - | - | - |
| F-ConvNet [37] | 0.904 | 0.724 | 0.848 | 0.825 |
| Regionlet [38] | 0.848 | 0.612 | 0.704 | 0.721 |
| DPM-VOC+VP [39] | 0.750 | 0.449 | 0.424 | 0.541 |
| 3DVP [40] | 0.875 | - | - | - |
| SubCat [41] | 0.841 | - | - | - |
| CompACT-Deep [42] | - | 0.587 | - | - |
| DeepParts [43] | - | 0.587 | - | - |
| Fast RCNN+VGG16 [9] | 0.860 | 0.625 | 0.688 | 0.724 |
| SCM-Multi (ours) | **0.939** | **0.804** | **0.900** | **0.881** |

Note: the top value under each metric is in **bold face**.

average precisions (MAP). It could be observed that our *SCM-Multi* network achieved top values on all these metrics.

According to these ablation evaluations and comparisons with other modern detectors on different datasets, it's shown that structural constraint mechanism is able to improve object detection quality on various network architectures, and is able to assist some prototype networks to achieve advanced performances.

### 4.3 Visualization analysis

We analyze behaviors of structural constraint mechanism during training and testing in this subsection. For this purpose, we visualized changing of the loss terms in structural constraint, their influences on feature space and some final detection results.

**Changing of loss values.** We plotted curves of Fisher loss and equi-proportion loss during training of object detection networks of different architectures. The observation subjects include RetinaNet, SSD, Faster RCNN and Cascade RCNN, all with structural constraints applied. These loss curves are shown in Fig 2. Both losses were obviously dropping during all these training processes. This observation indicates that the loss terms in structural constraints are effectively minimized, so they are indeed guiding networks' training.

**Influence on network feature space.** To observe the influences of structural constraint mechanism on object detection networks' feature spaces, we adopted t-SNE [44] to project high-dimensional backbone features to 2D space for visualization. These backbone features were obtained by feeding the networks with images of object classes. These images are sampled from `KITTI` according to its bounding box labels and are of class `Car` or `Pedestrian` (`Ped`). The extracted backbone features are then resized to a uniform size for the convenience of t-SNE transform. The visualization results are shown in Fig 3. The network subjects are Faster RCNN and Cascade RCNN. It could be observed that with greater extent of structural constraint application, the distributions of `Car` and `Ped` are less mixed and easier to separate.

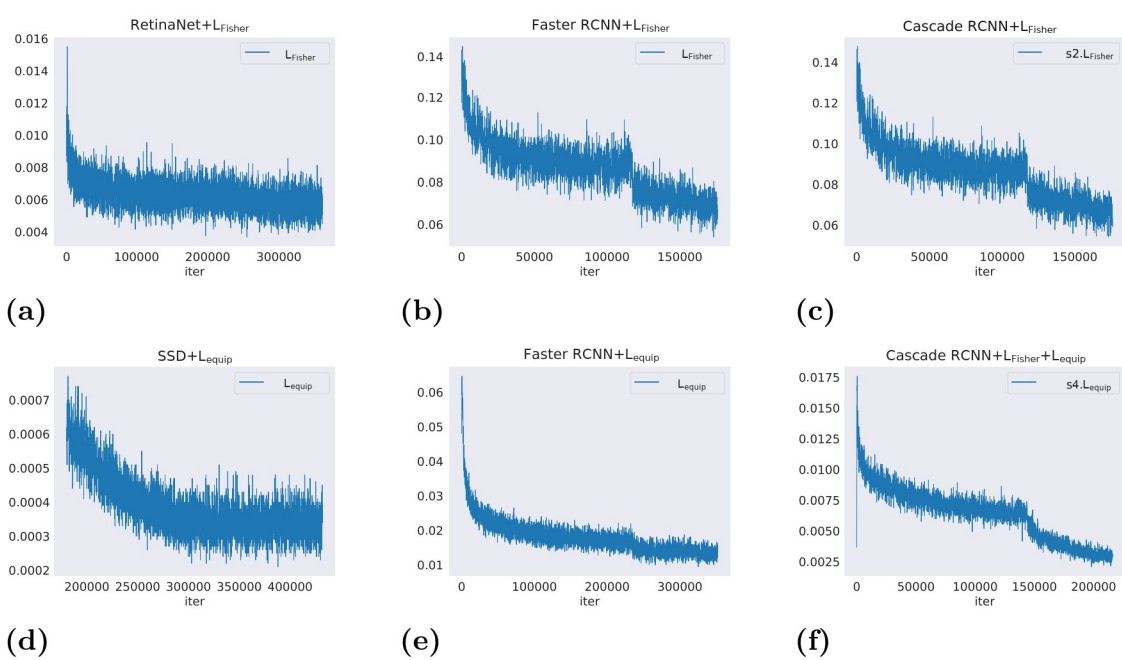

**Fig 2. The curves of Fisher and equi-proportion losses ($L_{Fisher}$ and $L_{equip}$) during the training of object detection networks of different architectures.** Upper row: Fisher losses; lower row: equi-proportion losses. "s#" in legends indicates the loss corresponds to stage # in the case of multi-stage networks.

This is a beneficial behavior to object classification, and is consistent with the intention of structural constraints.

**Detection result visualization.** In Fig 4, we visualized some detection results on `MSCOCO2017` images (`val2017`). We compared detection results of Faster RCNNs with

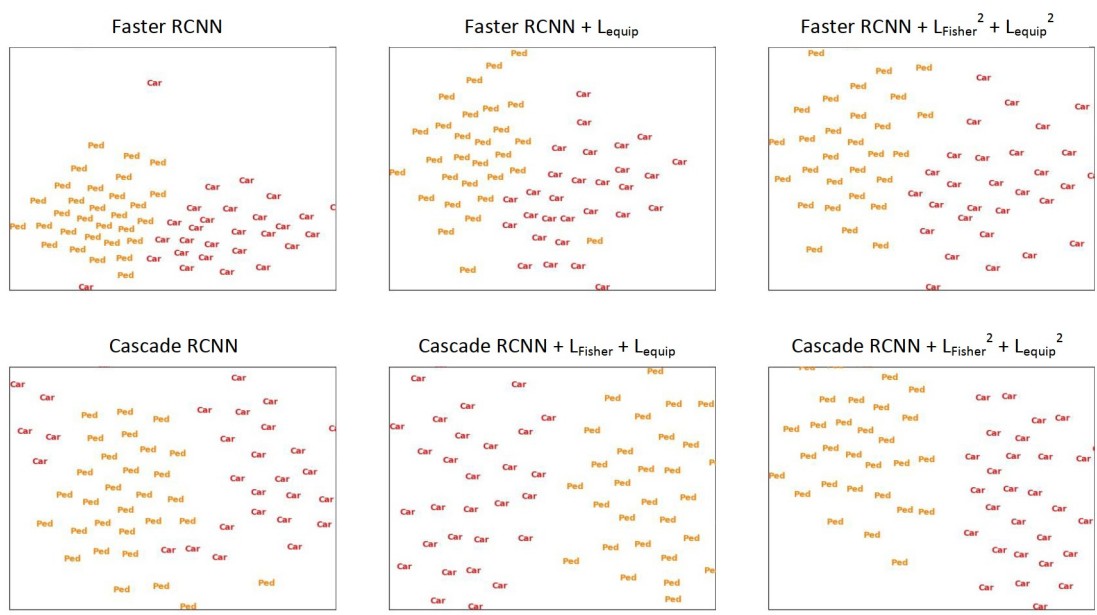

**Fig 3. t-SNE visualization of `Car` and `Pedestrian` (`Ped`) instance distributions in feature spaces of object detection networks with and without structural constraints applied.**

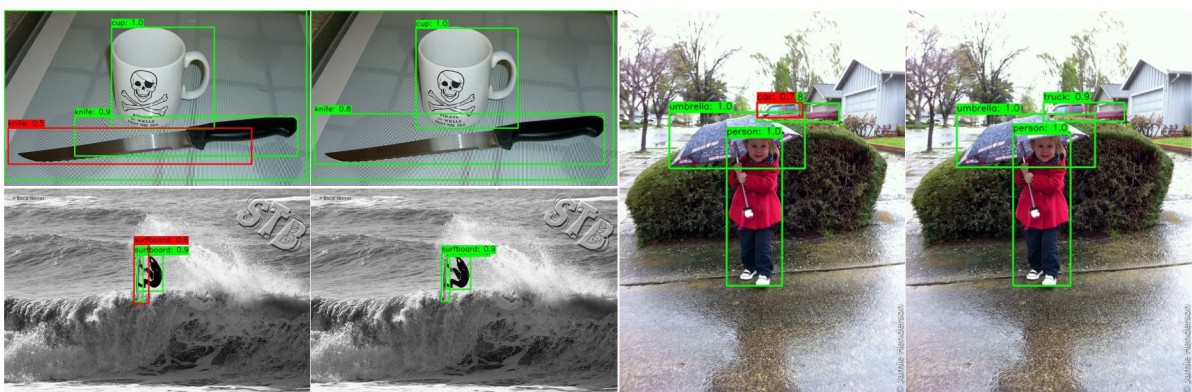

**Fig 4. Detection results of Faster RCNN (left in each couple) and Faster RCNN $+L^2_{Fisher} + L^2_{equip}$ (right in each couple).** Green boxes: correct detection results; red boxes: incorrect detection results. Each box is marked with its estimated class name and confidence score.

and without structural constraints applied. It could be observed that the application of structural constraints made the detector more accurate at localization and give less false positives.

## 5 Conclusion

In this work, we introduced our structural constraint mechanism for improving object detection quality. Structural constraint mechanism supervises object detection networks' intermediate feature spaces, and guides the training processes to optimize object class instances' distributions within the spaces. It constrains feature similarities of training sample pairs to be consistent with corresponding ground truth label similarities. With the aid of proxy feature design, structural constraint could be applied to all types of object detection network architectures. Experiment results indicate our structural constraint mechanism is able to optimize networks' intermediate features and consequently final detection results. It should be pointed out that calculation of structural constraint is done for all possible pairs of training samples, which has high GPU memory demand. We will address this issue in our future work.

## Author Contributions

**Conceptualization:** Zihao Rong.

**Data curation:** Zihao Rong.

**Funding acquisition:** Dehui Kong.

**Methodology:** Shaofan Wang.

**Project administration:** Dehui Kong.

**Resources:** Dehui Kong, Baocai Yin.

**Software:** Zihao Rong.

**Supervision:** Shaofan Wang, Dehui Kong.

**Validation:** Shaofan Wang.

**Visualization:** Zihao Rong.

**Writing – original draft:** Zihao Rong.

**Writing – review & editing:** Shaofan Wang.

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
