## [Decision Letter · Decision Letter 0]

11 Feb 2022

PONE-D-21-39942Improving object detection quality with structural constraintsPLOS ONE

Dear Dr. Rong,

Thank you for submitting your manuscript to PLOS ONE. After careful consideration, we feel that it has merit but does not fully meet PLOS ONE’s publication criteria as it currently stands. Therefore, we invite you to submit a revised version of the manuscript that addresses the points raised during the review process. Please Revise the paper by considering the reviewers' comments. Please submit your revised manuscript by Mar 28 2022 11:59PM. If you will need more time than this to complete your revisions, please reply to this message or contact the journal office at plosone@plos.org. Please include the following items when submitting your revised manuscript:A rebuttal letter that responds to each point raised by the academic editor and reviewer(s). You should upload this letter as a separate file labeled 'Response to Reviewers'.A marked-up copy of your manuscript that highlights changes made to the original version. You should upload this as a separate file labeled 'Revised Manuscript with Track Changes'.An unmarked version of your revised paper without tracked changes. You should upload this as a separate file labeled 'Manuscript'.

We look forward to receiving your revised manuscript.

Kind regards,

Jie Zhang

Academic Editor

PLOS ONE

Journal Requirements:

4. We note you have included a table to which you do not refer in the text of your manuscript. Please ensure that you refer to Table 1, 2, 3 and 4 in your text; if accepted, production will need this reference to link the reader to the Table.

Reviewers' comments:

Reviewer's Responses to Questions

**Comments to the Author**

1. Is the manuscript technically sound, and do the data support the conclusions?

Reviewer #1: Yes

Reviewer #2: Yes

2. Has the statistical analysis been performed appropriately and rigorously? 

Reviewer #1: Yes

Reviewer #2: Yes

3. Have the authors made all data underlying the findings in their manuscript fully available?

Reviewer #1: Yes

Reviewer #2: Yes

4. Is the manuscript presented in an intelligible fashion and written in standard English?

Reviewer #1: Yes

Reviewer #2: Yes

5. Review Comments to the Author

Reviewer #1: This paper proposes an approach that utilizes Structural Constraints to improve performance of object detection tasks. The proposed Structural Constraints measurement consists of two parts: Fisher loss and Equi-proportion loss. The fisher loss is helpful in improving classification performance, and the equip loss function is designed for improving localization performance. Finally, a series of strategies are proposed to apply the Structural Constraints into different frameworks of object detection. Extensive experiments are conducted to demonstrate the effectiveness of proposed approach.

1) Positive Points.

Most of the modern object detection networks constrain only classification loss and localization loss, and this paper states that these losses are not enough and then further utilizes the Fisher loss and the Equip loss. Such a solution is interesting and could improve performance of object detection. In addition, the presentation is clear and the paper is easy to read.

2) Negative Points.

The main contribution of this paper is the Structural Constraints that introduce the Fisher loss and the Equip loss. However, the experiments are insufficient to verify the positive effects of the two losses. The ablation experiments on RetinaNet some scores become even worse after using the Fisher loss (Table 1 and Table 2), making it somehow useless for the one-stage object detection framework. In addition, the ablation experiments on Cascade RCNN* show that the Equip loss only improves the AR indexes but becomes harmful for the AP indexes in most cases, implying that the Equip loss is not useful for the multi-stage object detection network. The authors should explain and further justify the positive effects of these two losses.

Another concern is that the overall performance is not very satisfactory (i.e., not state-of-the-art). Many superior object detection networks have been proposed in recent years (e.g., DETR). Although the proposed approach claims that their solution can be applied to all object detection network architectures, more comparative experiments on recent object detection works should be done to demonstrate the effect of proposed approach.

Reviewer #2: Object detection is an important research direction in computer vision and the basis of many downstream work.

In this paper, the author investigated a large number of literatures and found that in object detection, in addition to the commonly used classification loss and regression loss, the addition of other loss functions can improve the detection effect, such as focal loss. At the same time, from the analysis of clustering and other tasks, it is found that the constraint based on the mutual relations between training samples can effectively improve feature learning. Therefore, the authors propose to use structural constraint mechanism in object detection, and improve the effect of object detection by adding structural constraint loss to object detection algorithm. The authors add Fisher loss to the classification branch of object detection and Equi-proportion loss to the location branch, and completes the corresponding training with the help of an proxy feature. Experiments on Object detection public data sets mscoco2017 and Kitti show the effectiveness of this method.

The main problems are as follows:

1、 In this paper, the two additional loss functions are directly added to the original loss function, such as formula 4, but in the following 3.3, it is explained that they are used in different branches, and they are inconsistent.

2、 After adding the loss function, how to train the network? Is it to train with the classification loss first, and then use the Fisher loss for fine-tuning? It needs to be explained in detail here.

3、 The super parameters in the experimental settings, such as the setting of IOU threshold, directly give specific values, but do not explain how to select this parameter value. Further clarification is required.

4、 What is the difference between the Fisher loss added in this paper and the Fisher discriminative layer in reference paper 2.

6. PLOS authors have the option to publish the peer review history of their article (what does this mean?). If published, this will include your full peer review and any attached files.

Reviewer #1: **Yes: **Jia Li

Reviewer #2: No

---

## [Author Response · Author response to Decision Letter 0]

23 Mar 2022

The responses to reviewers are uploaded as PDF files.

---

## [Decision Letter · Decision Letter 1]

18 Apr 2022

Improving object detection quality with structural constraints

PONE-D-21-39942R1

Dear Dr. Rong,

We’re pleased to inform you that your manuscript has been judged scientifically suitable for publication and will be formally accepted for publication once it meets all outstanding technical requirements.

Kind regards,

Jie Zhang

Academic Editor

PLOS ONE

Additional Editor Comments (optional):

Reviewers' comments:

Reviewer's Responses to Questions

**Comments to the Author**

1. If the authors have adequately addressed your comments raised in a previous round of review and you feel that this manuscript is now acceptable for publication, you may indicate that here to bypass the “Comments to the Author” section, enter your conflict of interest statement in the “Confidential to Editor” section, and submit your "Accept" recommendation.

Reviewer #1: All comments have been addressed

Reviewer #2: All comments have been addressed

2. Is the manuscript technically sound, and do the data support the conclusions?

Reviewer #1: Partly

Reviewer #2: Yes

3. Has the statistical analysis been performed appropriately and rigorously? 

Reviewer #1: No

Reviewer #2: Yes

4. Have the authors made all data underlying the findings in their manuscript fully available?

Reviewer #1: Yes

Reviewer #2: Yes

5. Is the manuscript presented in an intelligible fashion and written in standard English?

Reviewer #1: Yes

Reviewer #2: Yes

6. Review Comments to the Author

Reviewer #1: Most of my concerns have been well addressed. I think the paper can now be published in its current form.

Reviewer #2: This manuscript has been well improved, and I think all of my comments have been addressed with new analysis.

7. PLOS authors have the option to publish the peer review history of their article (what does this mean?). If published, this will include your full peer review and any attached files.

Reviewer #1: **Yes: **Jia Li

Reviewer #2: No

---

## [Editor Report · Acceptance letter]

9 May 2022

PONE-D-21-39942R1 

Improving object detection quality with structural constraints 

Dear Dr. Rong:

I'm pleased to inform you that your manuscript has been deemed suitable for publication in PLOS ONE. Congratulations! Your manuscript is now with our production department. 

Kind regards, 

on behalf of

Dr. Jie Zhang 

Academic Editor

PLOS ONE